# Edible/Biodegradable Packaging with the Addition of Spent Coffee Grounds Oil

**DOI:** 10.3390/foods12132626

**Published:** 2023-07-07

**Authors:** Dani Dordevic, Simona Dordevic, Fouad Ali Abdullah Abdullah, Tamara Mader, Nino Medimorec, Bohuslava Tremlova, Ivan Kushkevych

**Affiliations:** 1Department of Plant Origin Food Sciences, Faculty of Veterinary Hygiene and Ecology, University of Veterinary Sciences Brno, Palackého tř. 1946/1, 612 42 Brno, Czech Republicdordevics@vfu.cz (S.D.);; 2Department of Meat Hygiene and Technology, Faculty of Veterinary Hygiene and Ecology, University of Veterinary Sciences, 612 42 Brno, Czech Republic; 3Department of Medical Laboratory Technology, College of Health and Medical Techniques, Duhok Polytechnic University, Duhok 42001, Iraq; 4University North, Dr. Zarka Dolinar Square 1, 48000 Koprivnica, Croatia; 5Department of Experimental Biology, Faculty of Science, Masaryk University, 625 00 Brno, Czech Republic

**Keywords:** edible packaging, biodegradable packaging, coffee, waste, oil

## Abstract

Background: Following petroleum, coffee ranks as the second most extensively exchanged commodity worldwide. The definition of spent coffee ground (SCG) can be outlined as the waste generated after consuming coffee. The aims of the study are to produce edible/biodegradable packaging with the addition of spent coffee grounds (SCG) oil and to investigate how this fortification can affect chemical, textural, and solubility properties of experimentally produced films. Methods: The produced films were based on κ-carrageenan and pouring–drying techniques in petri dishes. Two types of emulsifiers were used: Tween 20 and Tween 80. The films were analyzed by antioxidant and textural analysis, and their solubility was also tested. Results: Edible/biodegradable packaging samples produced with the addition of SCG oil showed higher (*p* < 0.05) antioxidant capacity in comparison with control samples produced without the addition of SCG oil. The results of the research showed that the fortification of edible/biodegradable packaging with the addition of SCG oil changed significantly (*p* < 0.05) both chemical and physical properties of the films. Conclusions: Based on the findings obtained, it was indicated that films manufactured utilizing SCG oil possess considerable potential to serve as an effective and promising material for active food packaging purposes.

## 1. Introduction

After petroleum, coffee is the most traded commodity throughout the world [1]. The description of spent coffee ground (SCG) is the following: (i) it is the waste accumulated after coffee consumption; (ii) it is a byproduct of the brewing process. The fact is that coffee consumption has been increasing worldwide and, consequently, so has the accumulation of this byproduct. It is estimated that out of 1 g of coffee, 0.91 g represents a waste or spent coffee ground; to this estimation should be added that around 6 million tons of spent coffee ground is accumulated worldwide yearly [2,3]. Other literature data are stating that annual consumption of coffee beans is around 8 million tons [4]. This amount of waste accumulation is making spent coffee ground a big economic and environmental problem [5,6]. SCG has a high humidity and also a high organic load. The advantage of SCG is that it can be easily collected from cafeterias, restaurants, and factories. Moreover, SCG has a high humidity and organic load, making it a good substrate for the mold activity [7].

SCG contains, on average, 45.3% (calculated on *w*/*w* dry weight) of polysaccharides (mainly mannose, galactose, arabinose and glucose; these sugars are bound to cellulose and hemicellulose complexes). The major component of polysaccharides in SCG are mannans [8,9]. Reducing sugars are virtually absent in SCG [10]. Polyphenolic compounds represent an important factor for SCG antioxidant activity [11]. Chlorogenic acid is the main phenol found in SCG (accounting for up to 14%, based on dry matter) [12]. 

Spent coffee grounds contain up to 20 wt% of oil [13,14]. The main fatty acids in SCG are linoleic fatty acid (C18:2, over 40%) and palmitic fatty acid (C16:0, over 30%) [15]. 

A biodegradable polymer, polyhydroxybutyrate (PHB), which is an alternative to petro-based plastics, has been already produced with the use of SCG oil [4]. Waste disposal is also connected with the accumulation of food packaging, mainly produced as certain plastic material [16]. Edible and biodegradable packaging, produced from polysaccharides, represents the way leading to the reduction of non-biodegradable material used for food packaging [17]. Previous research and experiments emphasized that there are many ways and recipes for the production of edible/biodegradable packaging [18]. On the other hand, the production of plastic packaging is also influenced by consumers’ demand for more environmentally friendly packaging that also has properties that could increase shelf-life and safety of wrapped commodities [19].

It is common knowledge that lignocellulosic materials derived from natural resources, such as coffee grounds, are desirable candidates as additive or reinforcing agents for Poly(butylene adipate-co-terephthalate) (PBAT), not only to improve its mechanical properties but also to lower the costs of the finished product [20]. The torrefaction process, which consists of decreasing the water content of spent coffee ground, has been successfully used as a hydrophobic reinforcing agent for biodegradable PBAT [21], thus allowing for the generation of green composites without the need for a compatibilizer. Different typologies of fiber materials have been used for the improvement of polystyrene biocomposites, with the main identified disadvantage being the premature breaks in the polymer chains [22].

Active packaging (films and coatings) is developed with the addition of functional compounds that can extend the shelf-life of food with the aim of improving safety and quality of the food due to migration to the wrapped commodity [23]. Due to the presence of functional compounds in discarded agricultural and food waste, usage of such waste in the production of packaging has the potential to allow for a more sustainable and eco-friendlier production [24]. 

Coffee oil is mainly used in the cosmetic and pharmaceutical industry, serving as an emollient agent, antioxidant, effective skin hydrator, and UVB protector [19]. Coffee oil (from raw beans, roasted beans, and spent coffee ground) has an ultraviolet absorption property due to the main fatty acid, linoleic acid. The properties of coffee oil can vary slightly depending on the coffee cultivar as well as the extraction material used, including raw beans, roasted beans, and spent coffee grounds [11]. Triacylglycerols make up the majority of coffee’s lipid component (75%), along with a sizeable unsaponifiable portion (19%) that is rich in diterpenes (cafestol and kahweol) and contains sterols, tocopherols, phosphatides, and waxes (tryptamine derivatives) [25].

Spent coffee ground represents the product from which the oil can be extracted. It is believed that, since spent coffee ground contains bioactive compounds, such as polyphenolic compounds, the addition of SCG can affect the antioxidant properties of the product to which it is added. Therefore, the fortification of edible/biodegradable packaging can influence not only textural parameters, but also the antioxidant properties of both packaging and wrapped food due to the migration. Edible/biodegradable packaging can further serve as a bioindicator of food condition, mainly due to its color changing property [26,27,28].

The aim of the study was to incorporate the oil extracted from spent coffee ground to the edible/biodegradable packaging to investigate how the addition of this byproduct affects physical, chemical, and textural properties of experimentally produced packaging.

## 2. Materials and Methods

### 2.1. Edible/Biodegradable Packaging Production

For each prepared film, 0.3 g of κ-carrageenan was weighed. In films with the addition of coffee oil, 0.1, 0.45, 0.8, or 1 g of oil was weighed. Distilled water was added to the final volume of 45 mL. Samples were successively stirred until dissolution and placed on the hotplate, where they were heated until liquefaction. Samples were then placed on the magnetic stirrer (350 rpm, 50 °C) for 10 min and 0.25 mL of glycerol was subsequently added, followed by 5 min of stirring. At the end, 1.3 mL of Tween 20 or Tween 80 was added and samples were stirred for 7 min. Prepared film-forming solutions were poured in petri dishes with a diameter of 9 cm to dry. The ingredients used in the experimental production of edible/biodegradable packaging are shown in Table 1.

### 2.2. Oil Extraction from Spent Coffee Ground

Spent coffee ground was collected from coffee shops in Brno (the Czech Republic), and co-sited from the mixture of Arabica and Robusta cultivars (50:50). The oil was extracted via the following procedure: 200 g of coffee ground was put in 400 mL of hexan. The mixture was shaken several times and, after four days of storage, it was filtrated through filter paper. The solution was placed on the magnetic stirrer and stirred for 4 h (250 rpm) to evaporate the hexan. The obtained oil was put in the refrigerator.

### 2.3. Fat Content Determination in the Spent Coffee Ground (SCG)

Using the B-811 apparatus from Büchi in Flawil, Switzerland, the Soxhlet technique was used to determine the total fat content. The 5 g sample of used coffee grounds was extracted using petrol ether as the solvent. The following program was adhered to: 90 min of extraction, 30 min of washing, and 20 min of solvent evaporation.

### 2.4. Acid Value (AV) Determination of Oil Extracted from the Spent Coffee Ground

The acid value (AV) was assessed in accordance with ISO 660:2009. Following this, 50 mL of diethylether and 1 mL of 1% phenolphthalein was added to 5 g of the spent coffee ground sample. Following a minute of shaking, the sample was titrated with 0.1 M KOH.

### 2.5. Peroxide Value (PV) Determination of Oil Extracted from the Spent Coffee Ground

According to the ISO 3960:2017 standard, the peroxide value (PV) was calculated as follows: 5 g of the spent coffee ground sample was mixed with 30 mL of a solution made of glacial acetic acid and chloroform (2:3). The mixture was shaken for 1 min and the next step consisted of the addition of 30 mL of distilled water and 5 mL of a 1% starch solution. To titrate the sample, 0.01 M Na_2_S_2_O_3_ was used. In place of the sample, water was added to the blank sample during preparation.

### 2.6. Determination of Total Polyphenol Content

According to Tomadoni et al., 2016 [29], the Folin–Ciocalteau method was used to measure the total polyphenol content. The material was weighed out at 0.1 g, and 20 mL of a 1:1 ethanol–water solution was then added. Samples were extracted in an ultrasonic bath for 30 min, after which 1 mL was collected in a 25 mL volumetric flask. The sample was then mixed with 5.9 mL of 7.5% Na_2_CO_3_ and 5.9 mL of a 1:10 diluted Folin–Ciocalteau solution. Successively, the samples were incubated in the dark for 30 min. A blind sample (1 mL of distilled water was substituted for 1 mL of the sample) was used to test the absorbance at 765 nm. The outcomes were presented in milligrams of gallic acid per gram.

### 2.7. Ferric Reducing Antioxidant Power (FRAP)

According to the publication by Behbahani et al., 2017 [30], 0.1 g of sample was weighed out, 20 mL of ethanol–water mixture (1:1) added, and the samples were then sonicated in a water bath for 30 min in order to evaluate the antioxidant activity using the FRAP method. A working solution composed of vinegar buffer, TPTZ, and FeCl_3_ was then added to 180 μL of the extract, 300 μL of distilled water, and 3.6 mL of the extract in dark glass vials. The samples underwent an additional 8 min of dark incubation. At 593 nm, the absorbance was measured in comparison with a blind sample made up of working solution and distilled water. The calibration curve was created using Trolox, and the results were presented as μmol of Trolox per gram of sample.

### 2.8. 2,2-Diphenyl-1-picrylhydrazyl (DPPH)

According to Sivarooban et al., 2008 [31], the DPPH technique was used to measure the antioxidant activity. Twenty mL of a 1:1 ethanol–water solution was poured into dark glass vials containing 0.1 g of the material, which was then sonicated for 30 min before being filtered. After 30 min of dark incubation, 3 mL of the extract was combined with 1 mL of 0.1 mM DPPH solution. At 517 nm, a CECIL spectrophotometer was used to detect the absorbance.
DPPH (%) = [(Abs_DPPH_ − Abs_DPPH_)/Abs_DPPH_] × 100

### 2.9. 2,2′-Azinobis-(3-ethylbenzthiazolin-6-sulfonic Acid) (ABTS)

According to Thaipong et al. (2006) [32], the antioxidant activity was assessed using the ABTS technique. Twenty mL of ethanol–water solution (1:1) was added, 0.1 g of the sample was weighed out into opaque glass vials, and the samples were then sonicated for 30 min before being filtered. Ten mL of 0.007 M ABTS solution and 10 mL of 0.00245 M potassium persulphate solution was combined 12–16 h prior to the measurement. Before the measurement, the solution was diluted to ensure that the final absorbance at 735 nm was 0.7. Then, 20 L of the produced extract was combined with 1980 L of ABTS solution. The samples were incubated in the dark for 5 min before the absorbance at 735 nm was measured. The results were calculated according to the following formula:ABTS (%) = [(Abs_ABTS_ − Abs_sample_)/Abs_ABTS_] × 100

### 2.10. Cupric Ion Reducing Antioxidant Capacity (CUPRAC)

According to Apak et al., 2004 [33], the antioxidant activity was assessed using the CUPRAC technique. Twenty mL of a 1:1 ethanol–water solution was poured into dark glass vials containing 0.1 g of the material, which was then sonicated for 30 min before being filtered. Successively, 0.1 mL of ethanol-water (1:1), 1 mL of the extract, 1 mL of NH_4_Ac buffer pH = 7.0, 1 mL of 0.0075 M Neocuproin, 1 mL of 0.01 M Copper (II), and 1 mL of the mixture was mixed. After an hour of dark incubation, the samples were evaluated for absorbance at 450 nm in comparison with a blind sample. The calibration curve was created using Trolox, and the results were represented as mol of Trolox per gram of sample.

### 2.11. Malondialdehyde (MDA)

Thiobarbituric acid (TBA) was used to evaluate MDA, with some changes, in accordance with the findings of Khalifa et al. (2016) [34]. The 1.5 g sample was transferred to a centrifugal tube, 1 mL of EDTA was added, and the mixture was blended. Next, 5 mL of 0.8% BHT was added, and the substance was once more mixed. The mixture was mixed with a 10% TCA solution for 30 s at 10,000 rpm. The tube was homogenized and then centrifuged at 3500× *g* for 5 min at 4 °C. The lower layer was filtered, and the upper layer was removed from the tube. The 10 mL volumetric flask was filled with the filtrate, and the mark was filled with 10% TCA. Four mL was withdrawn from this flask, and 1 mL of TBA was added. This underwent a 90-min incubation period at 70 °C, much like the conventional MDA solutions. After incubation, everything was moved to the cold bath for 1–2 min, following which the samples were left to rest for 45 min at room temperature. A spectrophotometer was used to detect the absorption at 532 nm. The results were calculated using the standard calibration curve.

### 2.12. Textural Properties of Packaging

Using the ASTM international test method, ASTM D882-02, a TA.XT plus texture analyzer (Godalming, UK) was used to assess strength (MPa) and elasticity (%). The manufactured packets were sliced into rectangles of 1 × 5 cm, and each measurement was taken five times.

### 2.13. Water Content, Solubility, and Swelling Degree

The determination was made using a slightly modified version of the Souza et al. (2017) approach. The film samples were divided into 22 cm squares, and the weights were recorded as W1 using an analytical scale (KERN, Germany). The films were then heated to 105 °C in an oven (Ecoccel 55) for two hours before being weighed again (W2). Samples were then put in beakers with 25 mL of water, left at room temperature for 24 h, dried, and reweighed (W3). Next, samples were transferred to an oven for 24 h at 105 °C and then weighed (W4). Replicates (*n* = 6) were prepared for each sample. The solubility was measured in distilled water and in seawater collected in the vicinity of Gdańsk (Poland) at the following map coordinates: 54.342504, 19.028122.

The results were obtained from the following equations:Water content (%) = [(W − W_2_)/W_1_)] × 100
Solubility (%) = [(W_2_ − W_4_)/W_2_] × 100
Swelling degree (%) = [(W_3_ − W_2_)/W_2_] × 100

### 2.14. Statistical Analysis

To determine whether differences within groups were statistically significant at the level of 0.05, one-way ANOVA, the parametric Tukey post hoc test (in the case of the Levene’s test revealing equal variances, *p* > 0.05), and the nonparametric Games–Howel post hoc test (in the case of the Levene’s test revealing unequal variances, *p* < 0.05) were all used. The SPSS 20 statistical program (IBM Corporation, Armonk, NY, USA) was used to conduct statistical analyses. Overall differences among samples were checked by principal component analysis (PCA).

## 3. Results and Discussion

The extracted oil yield from spent coffee ground, oxidation level, and antioxidant properties of used oils in experimentally produced edible/biodegradable packaging are shown in Table 2.

The fat content in spent coffee grounds was below 5%, but literature suggests that oil content in SCG can reach 20% [2]. Certainly, the brewing and extraction techniques had an impact on the lipid content of SCG [35].

The oxidative stability of the extracted SCG oil was measured by the content of peroxides, as the primary products of the oxidation. The peroxide value (PV) in freshly extracted SCG oil was 7.13 ± 0.81 mekv. O_2_/kg. This number of primary products of oxidation cannot be considered high, since PV over 20 mekv. O_2_/kg indicates the rancid taste of oil [36]. 

Extracted SCG oil showed non-negligible antioxidant properties, the latter being comparable to the antioxidant properties of other extensively used cold-pressed vegetable oils such as rapeseed and olive oil [37,38]. 

According to previous research, it was found that microencapsulated samples with coffee oil had a higher antioxidant activity than samples without coffee oil. Previous experiments and research also included in vitro testing of SCG oil and showed good antioxidant activity and cytotoxic effects on skin and liver cells [39,40].

The antioxidant profile, based on antioxidant capacity and total polyphenol content, of experimentally produced edible/biodegradable packaging is shown in Table 3. The oxidation statute of edible/biodegradable packaging was measured with the content of malondialdehyde (MDA) and examined values are shown in Table 3.

Total phenol content (TPC) unambiguously (significantly *p* < 0.05) increased with a larger addition of SCG oil. The highest TPC was recorded in the sample CA1TW80 (7.00 ± 0.15 mg gallic acid/g), i.e., the sample with 1 mL addition of oil from SCG. The obtained results do not differ substantially from published results [19] where biofilms produced with 0.2% of coffee oil TPC ranged from 1.3 mg/g to 5.44 mg/g. 

The highest (*p* < 0.05) antioxidant activity measured by FRAP was also found in samples with 1 mL of SCG oil (CA1TW20: 7.22 ± 0.35 µmol/g and CA1TW80: 12.67 ± 1.94 µmol/g). The difference between the antioxidant properties of packaging produced with Tween 20 and Tween 80 is noticeable. The structural and functional properties of Tween 20 and Tween 80 are similar, having two nonionic moieties: one hydrophilic and one hydrophobic. The whole principal of Tween 20 and Tween 80 acting in aqueous solutions is that these molecules tend to self-aggregate, forming colloidal particles that are lipophilic on the inside and hydrophilic on the outside. Both Tween 20 and Tween 80 are widely recognized as safe, according to the European Medicines Agency and US Food and Drug Administration. Antioxidant activity measured by DPPH was higher (*p* < 0.05) in samples with Tween 80. The authors from previous studies [41] also found higher antioxidant activity in samples with Tween 80; however, authors further noticed that antioxidant activity of Tween 20 and Tween 80 was not meaningful in practice since antioxidant activity, both of Tween 20 and Tween 80, was not related to scavenge free radicals’ activity but rather to biological interactions [41]. Tween 20 and Tween 80, also called Polysorbate 20 and Polysorbate 80, have been found to oxidize rapidly during exposure to air at 40 °C during 7 weeks [42]. 

Certainly, the addition of SCG oil resulted in higher antioxidant properties of experimentally produced films since the oil extracted from spent coffee grounds possess antioxidant properties, containing phenolic compounds, tannins, caffeine, and other antioxidants [14]. Leow et al. (2021) confirmed in their study that the oil extracted from spent coffee grounds possess antioxidant properties; authors used a DPPH assay for the determination of antioxidant properties. Compared with oils produced from other waste materials, spent coffee grounds oil is considered less expensive, it has better stability (due to presence of antioxidants), and it also has a pleasant smell [13]. 

Malondialdehyde content increased with the addition of SCG oil. The highest malondialdehyde content was measured in the sample CA1TW20 (12.75 ± 0.03 µg/g). [43] found that extraction methods (Soxhlet and accelerated solvent extraction, using propanol as a solvent, and supercritical CO_2_ extraction methods) applied to spent coffee ground did not affect significantly malondialdehyde content, though it was lower than in our samples, 0.76 µg/g. Malondialdehyde is defined as a toxic product accumulated by reactive oxygen species, and is connected to lipid hydrolysis and oxidative rancidity [44]. Some studies are declaring that 1.0 mg/kg malondialdehyde is a threshold limit for product acceptance by consumers [45].

Since SCG is rich in bioactive compounds, mainly polyphenols, this fact makes it a good fortification compound in other edible and biodegradable matrices. The production and widespread use of petroleum-based plastics has certainly offered producers and consumers convenience, but it has also led to an addiction to this kind of packaging and material, leading to environmental issues. The addition of coffee waste, such as SCG oil, is fortifying the final product and at the same time is making SCG, a waste product, a valuable raw material. Literature supports the recognition of SCG oil as valuable in the production of biopolymers [46].

The solubility in both distilled water and seawater, water content, and swelling degree in seawater of experimentally produced edible/biodegradable packaging are reported in Table 4.

The solubility of produced edible/biodegradable packaging was 100% in distilled water, while the solubility in seawater was below 100%. It can be noticed that the solubility (in seawater) of edible/biodegradable packaging was higher for the samples prepared with the addition of SCG oil, in comparison with the control samples (CACO) produced without addition of SCG oil, though the solubility was affected by the emulsifier addition (both Tween 20 and Tween 80); the solubility in seawater was the lowest in the samples prepared with carrageenan and 0.1 g of SCG oil: 55.14 ± 0.36% (*p* < 0.05). It was observed that tween emulsifiers increase solubility, though in our study the linearity correlating with higher concentrations of Tween 20 and Tween 80 was not observed [47]. The seawater solubility of experimentally produced films in our research was not 100% after 24 h, but other researchers found that complete seawater dissolution of films prepared with carrageenan was achieved after seven days [48].

The solubility of the material can be increased not only by altering the composition or temperature, but also by manipulating the size of the particles themselves. It has been observed that smaller particle sizes, akin to the expanding dimensions of emulsion droplets, exhibit a direct correlation with the augmentation of solubility in a given substance [49].

The textural parameters (expressed as strength, breaking strain, and thickness) of experimentally produced edible/biodegradable packaging are presented in Table 5.

In comparison with films produced from chitosan and cinnamon oil, the samples produced in our research are showing a significantly higher thickness, since samples with cinnamon oil exhibited a thickness of 0.021 mm [50]. 

The strength of experimentally produced edible/biodegradable packaging was noticed to be lower (*p* < 0.05) in the samples produced with the addition of SCG oil and emulsifier (Tween 20 and Tween 80). SCG oil can act as a plasticizer and that is probably the reason for the strength decrease [51]. The authors [50] also noticed the decrease of tensile strength with the addition of cinnamon oils to the chitosan matric of films.

Certain PVC-based composites have been already produced with the use of SCG oil; these composites showed resistance to gamma radiation. The authors suggested that this property of PVC produced with SCG oil can be very useful for the packaging of materials that undergo gamma radiation sterilization [46].

The authors [19] observed that coffee oil addition below 0.2% to bioactive films of carboxymethyl cellulose did not yield significant changes in the mechanical properties of experimentally produced films. The oil derived from SCG was found to increase toughness when incorporated into polylactic acid (PLA)-based formulations due to the property of SCG oil to uniformly distribute within the polymeric matrix [52]. The addition of SCG oil resulted in increased interfacial adhesion, as well as in decreased oxygen permeability [53]. The breaking strain parameter increased (*p* < 0.05) with the addition of SCG oil. These results can be explained by the fact that packaging with SCG oil has a higher polyphenolic content since hydrogen bonds are often formed between polyphenols and polysaccharides [54]. The breaking strain is also connected to the amount of added glycerol, but all experimentally produced films with SCG oil contained the same amount of glycerol [50,55].

The tensile strength of all experimentally produced packaging was too low, but breaking strain was very high in comparison with properties of films produced by other authors [56]. The tensile strength was also low in comparison with results obtained with films with lapacho tea extracts, though the breaking strain values did not differ [57].

The thickness of packaging produced with SCG oil decreased significantly (*p* < 0.05) in comparison with control samples (CACO), ranging from 0.20 ± 0.02 mm (CA0.1TW80) to 0.66 ± 0.32 mm (CA0.1). Generally, edible films are produced with a thickness ranging from 0.010 mm to 0.100 mm [52].

Principal component analysis (PCA) is shown in Figure 1. All Antioxidant properties of experimentally produced edible/biodegradable packaging, taken into considerations by PCA, are shown in Figure 1A, while the overall properties estimated by PCA are shown in Figure 1B. 

Differences among groups are clearly identifiable in Figure 1, with the PCA showing significant differences between control samples (CACO) and packaging with the lowest SCG oil addition and no emulsifier addition (CA0.1). Antioxidant properties, as well as overall properties, were well affected by the amount of added SCG oil. According to the PCA analysis, overall differences were affected also by the typology of emulsifier. 

## 4. Conclusions

In this study, we successfully produced κ-carrageenan films by incorporating spent coffee grounds oil. The addition of spent coffee grounds oil had a significant impact on the antioxidant and physical properties of the edible/biodegradable packaging. Films with higher concentrations of spent coffee grounds oil exhibited increased polyphenol content and antioxidant activity, although the results were not entirely conclusive. The study highlighted how the choice of emulsifier greatly influenced the chemical and physical properties of the films. Films produced with Tween emulsifiers showed decreased solubility in seawater, reduced thickness, and increased breaking strain. These experimentally produced films have the potential for application with respect to various food products or even non-edible items, depending on the specific use. Further research should explore the utilization of the proposed edible/biodegradable packaging as an active packaging material for different food products, considering the films’ demonstrated higher antioxidant capacity.

## Figures and Tables

**Figure 1 foods-12-02626-f001:**
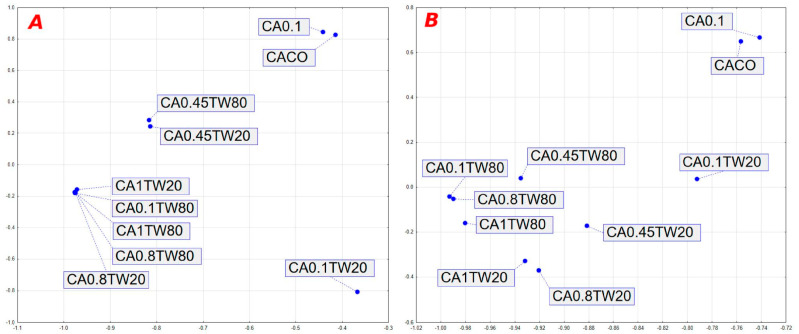
Principal components analysis (PCA) of edible/biodegradable packaging produced with the addition of spent coffee ground oil, antioxidant properties (**A**) and overall properties (**B**).

**Table 1 foods-12-02626-t001:** The sample labeling and used ingredients.

Sample Labeling	Ingredients
CACO	Control
CA0.1TW20	Carrageenan + 0.1 mL of oil from spent coffee ground + tween 20
CA0.45TW20	Carrageenan + 0.45 mL of oil from spent coffee ground + tween 20
CA0.8TW20	Carrageenan + 0.8 mL of oil from spent coffee ground + tween 20
CA1TW20	Carrageenan + 1 mL of oil from spent coffee ground + tween 20
CA0.1TW80	Carrageenan + 0.1 mL of oil from spent coffee ground + tween 80
CA0.45TW80	Carrageenan + 0.45 mL of oil from spent coffee ground + tween 80
CA0.8TW80	Carrageenan + 0.8 mL of oil from spent coffee ground + tween 80
CA1TW80	Carrageenan + 1 mL of oil from spent coffee ground + tween 80
CA0.1	Carrageenan + 0.1 mL of oil from spent coffee ground

**Table 2 foods-12-02626-t002:** Chemical properties of oil extracted from the spent coffee ground (SCG).

Analysis	Oil Extracted from SCG
Fat content (%)	3.59 ± 0.45%
Acid value (mg KOH/g)	8.4 ± 0.96
Peroxide value (mekv. O_2_/kg)	7.13 ± 0.81
FRAP Trolox (µmol/g)	5.37 ± 0.32
ABTS (%)	2.15 ± 0.24
Cuprac (Trolox µmol/g)	22.77 ± 0.79
Total phenolic content (mg gallic acid/g)	1.89 ± 0.14
Malondialdehyde (µg/g)	0.51 ± 0.00

**Table 3 foods-12-02626-t003:** Antioxidant profile of edible packaging prepared with the oil from spent coffee ground.

Samples	DPPH (%)	FRAP Trolox (µmol/g)	ABTS (%)	Cuprac Trolox (µmol/g)	TPC (mg gallic acid/g)	Malondialdehyde (µg/g)
CACO	7.61 ± 3.12 ^afg^	1.74 ± 0.24 ^agh^	2.27 ± 0.12 ^a^	1.84 ± 0.14 ^a^	0.54 ± 0.07 ^a^	0.43 ± 0.00 ^a^
CA0.1TW20	1.95 ± 0.44 ^cafg^	1.87 ± 0.20 ^ca^	2.76 ± 0.09 ^b^	5.11 ± 2.33 ^b^	2.23 ± 0.17 ^b^	9.20 ± 0.05 ^b^
CA0.45TW20	1.04 ± 0.06 ^dafg^	2.77 ± 0.19 ^da^	2.79 ± 0.13 ^b^	7.23 ± 0.25 ^ci^	2.87 ± 0.11 ^c^	11.58 ± 0.03 ^c^
CA0.8TW20	2.95 ± 1.28 ^dafg^	3.22 ± 0.18 ^ea^	3.16 ± 0.11 ^d^	5.37 ± 0.51 ^b^	2.91 ± 0.13 ^c^	8.88 ± 0.01 ^d^
CA1TW20	0.48 ± 0.20 ^f^	7.22 ± 0.35 ^f^	3.34 ± 0.14 ^dg^	10.19 ± 0.45 ^e^	3.80 ± 0.14 ^g^	12.75 ± 0.03 ^e^
CA0.1TW80	0.00 ± 0.00	0.88 ± 0.28 ^gc^	3.14 ± 0.17 ^d^	8.21 ± 0.68 ^c^	3.55 ± 0.14 ^f^	11.91 ± 0.01 ^f^
CA0.45TW80	7.37 ± 4.52 ^ga^	3.60 ± 0.21 ^hde^	3.46 ± 0.05 ^g^	7.40 ± 0.20 ^ic^	4.02 ± 0.09 ^g^	7.84 ± 0.01 ^g^
CA0.8TW80	13.74 ± 0.58 ^he^	9.22 ± 1.30 ^i^	3.70 ± 0.10 ^h^	6.46 ± 0.17 ^bi^	4.35 ± 0.11 ^e^	7.83 ± 0.02 ^g^
CA1TW80	7.65 ± 5.78 ^ba^	12.67 ± 1.94 ^j^	3.81 ± 0.07 ^h^	5.29 ± 0.28 ^b^	7.00 ± 0.15 ^i^	8.28 ± 0.00 ^i^
CA0.1	8.95 ± 1.85 ^eba^	2.91 ± 0.08 ^ah^	3.91 ± 0.11 ^h^	14.16 ± 1.28 ^k^	1.01 ± 0.14 ^j^	4.61 ± 0.02 ^j^

Values are expressed as mean ± standard deviation. Different letters in the same column indicate significant differences (*p* < 0.05).

**Table 4 foods-12-02626-t004:** The solubility of edible/biodegradable packaging samples in distilled water and seawater.

Samples	Solubility in Distilled Water (%)	Solubility in Seawater (%)	Swelling Degree in Seawater (%)	Water Content (%)
CACO	100	60.82 ± 2.75 ^a^	477.93 ± 18.22 ^a^	10.44 ± 1.37 ^a^
CA0.1TW20	100	87.95 ± 0.23 ^b^	148.27 ± 15.43 ^bce^	5.12 ± 0.40 ^b^
CA0.45TW20	100	85.30 ± 0.13 ^b^	127.99 ± 13.47 ^be^	6.80 ± 0.43 ^ab^
CA0.8TW20	100	77.70 ± 0.85 ^bce^	121.06 ± 9.15 ^b^	4.60 ± 0.52 ^b^
CA1TW20	100	71.99 ± 1.06 ^ea^	131.15 ± 14.37 ^be^	3.65± 0.09 ^b^
CA0.1TW80	100	83.39 ± 13.16 ^be^	197.70 ± 57.41 ^c^	4.58 ± 1.04 ^b^
CA0.45TW80	100	85.19 ± 0.79 ^b^	178.91 ± 7.76 ^bc^	4.99 ± 1.61 ^b^
CA0.8TW80	100	75.30 ± 0.36 ^cbe^	190.38 ± 14.52 ^ec^	5.73 ± 1.01 ^b^
CA1TW80	100	71.24 ± 2.23 ^cea^	163.53 ± 8.73 ^abc^	5.24 ± 0.40 ^b^
CA0.1	100	55.14 ± 0.36 ^d^	605.45 ± 17.53 ^d^	14.51 ± 3.31 ^c^

Values are expressed as mean ± standard deviation. Different letters in the same column indicate significant differences (*p* < 0.05).

**Table 5 foods-12-02626-t005:** Textural parameters of edible packaging prepared with the addition of spent coffee ground.

Samples	Strength (MPa)	Breaking Strain	Thickness (mm)
CACO	0.13 ± 0.02 ^a^	74.26 ± 2.30 ^a^	0.52 ± 0.08 ^ac^
CA0.1TW20	0.03 ± 0.01 ^be^	90.80 ± 4.43	0.29 ± 0.04 ^b^
CA0.45TW20	0.03 ± 0.02 ^db^	79.61 ± 19.00	0.29 ± 0.04 ^ab^
CA0.8TW20	0.04 ± 0.01 ^bce^	89.69 ± 14.93	0.37 ± 0.03 ^ab^
CA1TW20	0.04 ± 0.01 ^bce^	82.84 ± 3.81	0.34 ± 0.03 ^ab^
CA0.1TW80	0.06 ± 0.00 ^c^	95.72 ± 6.32 ^bc^	0.20 ± 0.02 ^b^
CA0.45TW80	0.06 ± 0.01 ^ec^	97.04 ± 3.58 ^b^	0.26 ± 0.02 ^b^
CA0.8TW80	0.05 ± 0.00 ^bce^	86.84 ± 7.80	0.31 ± 0.05 ^ab^
CA1TW80	0.05 ± 0.02 ^bce^	90.67 ± 16.58	0.30 ± 0.04 ^ab^
CA0.1	0.15 ± 0.02 ^a^	76.01 ± 2.49 ^ca^	0.66 ± 0.32 ^c^

Values are expressed as mean ± standard deviation. Different letters in the same column indicate significant differences (*p* < 0.05).

## Data Availability

All data underlying the study are available from the corresponding authors on reasonable request.

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
