# Peer review of "Edible/Biodegradable Packaging with the Addition of Spent Coffee Grounds Oil"

_foods, 2023, doi:10.3390/foods12132626_

Round 1

Reviewer 1 Report

Dear authors,

the present study is interesting, especially in the context of bioeconomy.

Line 91: coffee oil - please specify if commercial or oil extraction from spent coffee ground as described at 2.2.

In Table 1 you are referring to "x ml spent coffee ground" but according to the aim of your study: the aim of the study was to incorporate the oil extracted from spent coffee ground (line 85), therefore it is not clear the concept of your study and results obtained.

κ-carrageenan films with the addition of spent coffee grounds oil were successfully experimentally produced?!? Please clarify: did you use the oil extracted from spent coffee ground or did you use a certain quantity (ml) of spent coffee ground??? How much than oil has been in for example 1ml of spent coffee ground? please check Table 1.

Line 366: ....though not fully unambiguously - what do you mean?

The conclusions section needs to be reformulated, it is hard to understand.

Major spelling

Author Response

Dear Reviewer,
thank you very much for your time and positive evaluation of our manuscript. We would especially thank you for your positive recommendations and comments. We have improved our manuscript according to your important comments.

Best wishes, 

Ivan Kushkevych, corresponding author and my co-authors.

Our responses are below:

Dear authors,

the present study is interesting, especially in the context of bioeconomy.

Line 91: coffee oil - please specify if commercial or oil extraction from spent coffee ground as described at 2.2.

It was specified.

In Table 1 you are referring to "x ml spent coffee ground" but according to the aim of your study: the aim of the study was to incorporate the oil extracted from spent coffee ground (line 85), therefore it is not clear the concept of your study and results obtained.

We apologize for this mistake, it was revised.

κ-carrageenan films with the addition of spent coffee grounds oil were successfully experimentally produced?!? Please clarify: did you use the oil extracted from spent coffee ground or did you use a certain quantity (ml) of spent coffee ground??? How much than oil has been in for example 1ml of spent coffee ground? please check Table 1.

We apologize for this mistake, it was revised. We used the oil extracted from the spent coffee ground.

Line 366: ....though not fully unambiguously - what do you mean?

We wanted to stress out that not all samples showed significant increase in antioxidant capacity and polyphenols content with the increase addition of spent coffee ground oil.

The conclusions section needs to be reformulated, it is hard to understand.

We reformulated the whole conclusion part.

We would like to express our thankfulness to reviewer’s comments and suggestions.

With kind regards,
Authors

Reviewer 2 Report

Kindly see the report

Kindly see the report

Author Response

Dear Reviewer,

we appreciate a lot your time and expertise. We definitely improved the quality of our work with the application of your suggestions. We included useful references, but unfortunately FTIR analysis for this research we did not conduct. In our next work we will use certainly FTIR analysis since with the present work we gained knowledge as the base for the future work with especially spent coffee ground.

With kind regards,
Authors.

Round 2

Reviewer 1 Report

Authors addressed comments/suggestions. Manuscript could be accepted.

Minor spelling